# Arf GTPases Are Required for the Establishment of the Pre-Assembly Compartment in the Early Phase of Cytomegalovirus Infection

**DOI:** 10.3390/life11080867

**Published:** 2021-08-23

**Authors:** Valentino Pavišić, Hana Mahmutefendić Lučin, Gordana Blagojević Zagorac, Pero Lučin

**Affiliations:** 1Department of Physiology and Immunology, Faculty of Medicine, University of Rijeka, 51000 Rijeka, Croatia; valentino.pavisic@uniri.hr (V.P.); hana.mahmutefendic@uniri.hr (H.M.L.); plucin@unin.hr (P.L.); 2Nursing Department, University North, University Center Varaždin, Jurja Križanića 31b, 42000 Varaždin, Croatia

**Keywords:** Arf GTPases, cytomegaloviruses, pre-assembly compartment, Rab10, m06

## Abstract

Shortly after entering the cells, cytomegaloviruses (CMVs) initiate massive reorganization of cellular endocytic and secretory pathways, which results in the forming of the cytoplasmic virion assembly compartment (AC). We have previously shown that the formation of AC in murine CMV- (MCMV) infected cells begins in the early phase of infection (at 4–6 hpi) with the pre-AC establishment. Pre-AC comprises membranes derived from the endosomal recycling compartment, early endosomes, and the trans-Golgi network, which is surrounded by fragmented Golgi cisterns. To explore the importance of Arf GTPases in the biogenesis of the pre-AC, we infected Balb 3T3 cells with MCMV and analyzed the expression and intracellular localization of Arf proteins in the early phases (up to 16 hpi) of infection and the development of pre-AC in cells with a knockdown of Arf protein expression by small interfering RNAs (siRNAs). Herein, we show that even in the early phase, MCMVs cause massive reorganization of the Arf system of the host cells and induce the over-recruitment of Arf proteins onto the membranes of pre-AC. Knockdown of Arf1, Arf3, Arf4, or Arf6 impaired the establishment of pre-AC. However, the knockdown of Arf1 and Arf6 also abolished the establishment of infection. Our study demonstrates that Arf GTPases are required for different steps of early cytomegalovirus infection, including the establishment of the pre-AC.

## 1. Introduction

Cytomegaloviruses (CMVs) are widespread double-stranded DNA betaherpesviruses that mostly cause asymptomatic infection followed by a long-lasting latent state. In immunocompromised individuals, acute infection and reactivation from this latency can lead to life-threatening conditions [1,2,3]. CMVs virions are composed of dsDNA encased in capsid surrounded by tegument and envelope glycoproteins. Murine CMV (MCMV) shares many similarities with other betaherpesviruses and is used as a model for studying betaherpesvirus biology in vitro and in vivo. MCMV, upon entry, activates a gene-expression program, which is executed through three phases: immediate–early (IE), which lasts up to 2 h post-infection (hpi), early (E; from 2 to 16 hpi), and late (L; after 16 hpi) [4].

In the early stages of infection, CMV induces the reorganization of intracellular membranes and hijacks many cellular proteins as well as endocytic and secretory pathways of the host cells to form the virion assembly compartment (AC). The AC is a large and complex structure that is developed from the reorganized membranous organelles of the cell [5]. The onset of AC formation in murine MCMV-infected cells can be identified at 4–6 hpi as an extensive reorganization of the Golgi and the endosomal system [6,7]. A similar set of events was executed during HCMV infection with significantly delayed kinetics [8,9,10,11].

Studies on HCMV have demonstrated the alteration of expression of a large number of cellular genes and proteins that organize membranous organelle systems, and studies on both HCMV and MCMV indicate the extensive reorganization of early endosomes (EEs), the endosomal recycling compartment (ERC), and the trans-Golgi (TGN) related organelles [7,8,9,10,12]. The first reorganized structure observed at 6 hpi evolves throughout the early phase, which in MCMV infected cells lasts up to 15–16 h after infection. The end of the early phase is denoted by the initiation of viral DNA replication followed by the expression of a large set of MCMV-encoded gene products, known as late genes. Many of these gene products embed into the reorganized membranous organelles and further drive the maturation of the AC. In general, we consider the reorganized structures that involve the Golgi and EE-ER-TGN-derived organelles to be present during the early phase of infection as pre-AC, whereas these structures that are introduced upon the integration of the late-gene products as AC [6].

One of the hallmarks of reorganized membranous organelles in the pre-AC is extensive tubulation. The tubulation in the membranous system of uninfected cells is driven by small GTPases from the Ras family, especially members of the Arf subfamily. Arf proteins are small (~20 kDa) monomeric GTPases that control intracellular trafficking within the secretory [13,14,15] and endocytic [16,17,18] pathways. Arf proteins, similar to other small GTPases, function as molecular switches and cycle between their inactive (GDP-bound) and active (GTP-bound) form [13,14,16]. The activation of Arfs is spatially and temporally regulated by guanine nucleotide exchange factors (GEFs), proteins that facilitate the exchange of GDP to GTP, and GTPase activating proteins (GAPs), which facilitate the exchange of GTP to GDP. Activated Arf proteins bind to membranes and recruit different effector proteins, thereby regulating downstream processes [19,20].

Based on their sequence homology, Arfs are divided into class I (Arf1, Arf2, and Arf3), class II (Arf4 and Arf5), and class III (Arf6) [13,14]. Murine and human cells do not express Arf2 [21] Class I and II predominantly act in the secretory pathway (the endoplasmic reticulum and the Golgi), where they regulate vesicle budding by recruiting coat complexes [13,14,15,22,23], but they also act in the endosomal system and trans-Golgi network (TGN) [24,25,26,27,28]. Arf1 and Arf3 also regulate transport between GA and TGN [29,30]. In the endosomal system, Arf1 regulates the endocytosis of GPI-anchored proteins [31], Arf1 and Arf4 regulate the retrograde transport of recycling endosomes to the TGN [25], while Arf1 and Arf3 control recycling from the early stage and recycling endosomes to the PM [24]. Arf6 is the only member of the Arf family that does not play a part in the secretory pathway [13,14,17,32,33] but regulates various processes within the endosomal system, including endocytosis [26,34,35,36,37], actin cytoskeleton dynamics [38,39,40], endosomal recycling [41,42,43,44], and organization of the ERC [17,45]. Many downstream effector functions of Arf proteins are associated with the tubulation of membranous organelles and the development of transport carriers.

The role of Arf proteins has been studied in the pathogenesis of several bacterial and viral infections [39,46,47,48,49,50,51], while their role in the pathogenesis of CMV infection has been understudied. A study of human CMV infection suggested impaired Arf6 function associated with the reprogramming of endosomal trafficking [52]. Our recent studies on murine CMV (MCMV) demonstrated the over-recruitment of all Arf isoforms within the AC in the late phase of infection, at 48 hpi [6]. However, the dynamics of Arf expression during the MCMV infection cycle, intracellular localization in the early phase of infection, and their role in pre-AC formation have not been examined in detail.

Therefore, this study aimed to analyze the contribution of class I, II, and III Arf proteins in the biogenesis of the pre-AC membranous organelle reorganization events during the early phase of infection. We analyzed the expression level of all Arf isoforms and their localization during the early phase of MCMV infection. A much shorter duration of the early phase during MCMV infection enabled systematic analysis of early phase events under the conditions of the long-term knockdown of all Arfs by small interfering RNAs (siRNA silencing). Our study demonstrates the overexpression of class I and II Arfs, the over-recruitment of all Arfs at membranes of the pre-AC, and the requirement of Arf1, Arf3, Arf4, and Arf6 for the development of the pre-AC. However, our study also revealed that the contribution of Arf1 and Arf6 is most likely associated with the earliest stages of infection, either during virion entry or during the establishment of infection. Given that many processes of the AC biogenesis are conserved among beta-herpesviruses, our data may contribute to understanding the biogenesis of the HCMV assembly compartment and to identifying the host cell processes that may be targeted for the development of antivirals.

## 2. Materials and Methods

### 2.1. Cell Lines, Viruses, and Infection Conditions

All experiments were performed on Balb 3T3 fibroblasts that were obtained from American Type Culture Collection (ATCC, clone A31, ATCC CCL-163). Cells were cultured in Dulbecco’s modified Eagle’s medium supplemented with 10% fetal bovine serum (FBS), 2 mM L-glutamine, 100 mg/mL of streptomycin, and 100 U/mL penicillin (Gibco/Invitrogen, Grand Island, NY, USA) at 37 °C in 5% CO_2_.

The recombinant virus Δm138-MCMV (MC95.15) with the deletion of the fcr1 (m138) gene [53] was regularly used for infection. Cells were infected at a multiplicity of infection (MOI) of 5–10 with infectivity enhancment by centrifugation [54]. Detection of the immediate–early 1 (IE1) protein was used to determine the effectiveness of the infection [7]. In some experiments, the cells were infected at the same MOI with C3X-GFP MCMV (a kind gift from M. Messerle), in which a GFP expression cassette was inserted in front of the *ie2* gene [55] since it was previously shown that foreign genes could be inserted at this location without affecting the growth of the recombinant MCMV.

### 2.2. Antibodies

Rabbit polyclonal anti-Arf1, rabbit polyclonal anti-Arf3, and rabbit polyclonal anti-Arf5 antibodies were purchased from Abcam (Cambridge, UK); monoclonal mouse anti-Arf3 and anti-GM-130 antibodies were from BD Transduction Laboratories (San Jose, CA, USA); rabbit polyclonal anti-Arf4 and mouse monoclonal anti-Arf5 antibodies were purchased from LSBio (Seattle, WA, USA); rabbit monoclonal anti-Arf6 and rabbit monoclonal anti-Rab10 antibodies were from Cell Signalling (Danvers, MA, USA); and mouse anti-β-actin was from Millipore (Billerica, MA, USA). The MAbs to murine transferrin receptor (TfR) (clone R17 217.1.3) was used as a hybridoma culture supernatant purified by affinity chromatography. Mouse monoclonal antibodies to MCMV proteins IE1 (clone IE1.01 and clone CRO101) and m06 (clone CROMA229) were produced and validated by the University of Rijeka Center for Proteomics. Alexa Fluor (AF)^488^- and AF^555^-conjugated secondary antibody reagents to mouse IgG_2a_, mouse IgG_1_, rat IgG, and rabbit IgG were from Molecular Probes (Leiden, Netherlands), and AF^680^-conjugated IgG1 and IgG_2a_ as well as peroxidase-conjugated secondary reagents to mouse and rabbit IgG, were from Jacksons Laboratory (Bar Harbor, ME, USA).

### 2.3. siRNA Silencing

FlexiTube small interfering RNA (siRNA) to Arf1 (GS11840), Arf3 (GS11842), Arf4 (GS11843), Arf6 (GS11845) as well as a non-targeting negative control siRNA (1022076) were purchased from Qiagen (Hilden, Germany), and the siRNA of Arf5 (4390771) was purchased from Ambion (Berlin, Germany). Cells were transfected with the siRNAs using RNAiMAX Lipofectamine reagent (Invitrogen, Carlsbad, CA, USA) according to manufacturer guidelines, with the final siRNAconcentration being 20 nM unless otherwise indicated. The cells proceeded to experimental procedure 72 h after transfection. Transfection specificity and efficiency were monitored by Western blot and immunofluorescence microscopy.

### 2.4. Immunofluorescence and Confocal Analysis

Cells grown on coverslips were fixed for 20 min with 4% PFA at RT and were permeabilized for 20 min with 0.5% Tween 20 at 37 °C. After permeabilization, the cells were incubated with primary Abs for 60–90 min at RT, the unbound Abs were washed with PBS, and the cells were incubated for 60 min with an appropriate fluorochrome-conjugated secondary reagent. After three washes in PBS, the cells were embedded in Mowiol (Fluka Chemicals, Selzee, Germany)—DABCO (Sigma Chemical Co, Steinheim, Germany) in PBS containing 50% glycerol and were analyzed using confocal microscopy.

Imaging was performed on an Olympus Fluoview FV300 confocal microscope (Olympus Optical Co., Tokyo, Japan) equipped with Ar488, He/Ne 543, and He/Ne 633 lasers. The images were acquired using Fluoview software, version 4.3 FV 300 (Olympus Optical Co., Tokyo, Japan), PLAPO60xO objective, appropriate filters, and PMT detectors. The z-series of 0.5 μm optical sections were acquired sequentially with medium scan speed (1.65 s/scan). Images (515 × 512 pixels) were captured at different zoom values (zoom factor: 0.75–6.0) with pixel sizes from 481.47 nm × 481.47 nm to 60.18 nm × 60.18 nm. Confocal images were exported in a TIFF format and were analyzed using ImageJ 1.53c software. Focus plane images were used for image presentation and colocalization presentation by plotting profiles along the line. Borders of cells and nuclei were determined through overlapping immunofluorescence and transmission light microscope images.

Colocalization was quantitatively evaluated on images with a pixel size of 60.18 nm × 60.18 nm and 120.37 nm × 120.37 nm using the JACoP plugin [56] to calculate Manders’ overlap coefficients. The best-fit lower threshold to eliminate most of the signal background (Costes automatic thresholding method) was determined using the threshold tool and was confirmed by visual inspection. Measures were made on the entire z-series of 9–13 cells from 2–3 independent experiments.

The percentages of cells with juxtanuclear Rab10 accumulation, nuclear IE1, or cytoplasmatic m06 expressions were determined directly during microscopy (Olympus Fluoview FV300 confocal microscope, PLAPO60xO objective) and on images captured with the zoom factors of 0.75 and 1.5. At least 10 fields of view (~8–30 cells/field) were analyzed from each of the two to three independent experiments.

### 2.5. Quantification of Infection by Flow Cytometry

Uninfected or C3X-GFP MCMV-infected cells were collected through short trypsin treatment, washed in PBS containing 10 mM EDTA, HEPES pH = 7.2, 0.1% NaN3, and 2% FCS (PBS-A), and the GFP fluorescent signal was analyzed by means of flow cytometry using a FACSCalibur flow cytometer (Becton Dickinson & Co, San Jose, CA, USA). Dead cells were excluded using propidium iodide (1 µg/mL), and 10,000 viable cells were acquired. The fluorescence signal was determined as the mean fluorescence intensity (MFI) after subtracting the background fluorescence (ΔMFI) determined in the uninfected cells. The percentage of infected cells was calculated after thresholding on cells infected with C3X-GFP MCMV at 0 hpi.

### 2.6. Western Blot Analysis

RIPA lysis buffer supplemented with protease and phosphatase inhibitors was used for the preparation of cellular extracts for WB analysis. Samples were separated by SDS-PAGE and were blotted onto a polyvinylidene difluoride (PVDF-P) WB membrane (Millipore, Billerica, MA, USA) at 60 to 70 V for 1 h. Membranes were incubated with 1% blocking reagent (Roche Diagnostics GmbH, Mannheim, Germany) for 1 h. After three cycles of washing with 0.5% TBS-T buffer (pH7), the membranes were incubated with a peroxidase-conjugated secondary reagent diluted in TBS buffer containing 0.5% blocking reagent for 45 min. After being washed three times with TBS-T buffer, the membranes were incubated for 1 min with ECL Prime substrate (GE Healthcare, Chicago, IL, USA), and signals were detected by Transilluminator Alliance 4.7 (Uvitec Ltd., Cambridge, UK).

Western blot signals were quantified by using ImageJ software according to published protocols [57], and signals were normalized to associated actin control. Briefly, the normalization factor for every lane was calculated as a ratio between the observed actin signal for every lane and the highest observed signal of the actin for the blot. Normalized experimental signals were calculated by dividing the observed experimental signal by the lane normalization factor.

### 2.7. Statistics

The data are presented as means ± standard deviation. The significance of difference was tested using a two-tailed Student’s *t*-test, and differences were considered to be significant if *p* was <0.005. The asterisk above the error bars indicates statistical significance using the group transfected with non-targeting, scramble (SCR.) siRNA as the control group.

## 3. Results

### 3.1. Increased Expression of Class I Arf Proteins and Their Accumulation in the Pericentriolar Region of MCMV-Infected Cells at the Time of the Pre-AC Establishment

Our previous study demonstrated the accumulation of Arf (Arf1–6) proteins at the AC in the late phase of MCMV infection (at 48 hpi) [6]. Thus, we first analyzed the expression level and intracellular recruitment of all Arf proteins in the early phase of infection at the pre-AC stage. Considering that the reorganization of the membranous system and the development of the pre-AC in MCMV-infected cells is initiated already 5–6 h after infection [7,12,58], we analyzed their expression level by Western blot throughout the entire early phase of infection, up to 16 hpi, which is when viral DNA synthesis begins [59]. The intracellular localization was analyzed by double immunofluorescence staining and confocal microscopy at 6 hpi, a time when pre-AC rearrangement was observed in approximately half of the infected cells, and at 16 hpi, at the end of the early phase when, ~90% of the infected cells establish full pre-AC [7,12]. To avoid the unspecific capture of antibody reagents by a MCMV protein with a Fc receptor property, we infected cells with a recombinant virus Δm138-MCMV. MCMV-encoded immediate–early 1 (IE1) protein expression, which localizes in the nucleus of infected cells, was used to control infection.

We first analyzed the class I Arf proteins represented by Arf1 and Arf3 in murine cells. As demonstrated in Figure 1, both the Arf1 and Arf3 proteins were upregulated in MCMV-infected cells. The increased amount of Arf1 was already detected 2 h post-infection (hpi), as demonstrated by the representative Western blot (Figure 1A), and it was maintained throughout the entire early phase, up to the 16th hour of infection. The intracellular amount of Arf1 was approximately doubled, as determined by quantification of the Arf1 signal relative to the actin from the same samples (Figure 1B). A similar pattern was observed for Arf3 in the early phase of infection (Figure 1D,E). These data indicate that class I Arf proteins are rapidly upregulated in MCMV-infected cells, which can be a consequence of either increased synthesis or slowed degradation. The rapidity of upregulation together with the analysis of the host cell transcriptome, which demonstrated no significant alteration in any of the mRNAs of the Arf proteins in the early phase of infection [6], suggest that observed increases of the levels of Arf proteins are more likely due to altered mechanisms controlling endogenous class I Arf protein degradation rather than a consequence of altered synthesis.

The endogenous expression level and membrane recruitment of Arf1 (Figure 1C) and Arf3 (Figure 1F) was relatively low in Balb 3T3 cells immediately after infection (0 hpi), which was similar to the uninfected cells [6]. Thus, we further examined whether the increased protein level was associated with the increased recruitment at the reorganized membranes of the pre-AC. As demonstrated in Figure 1C,F, both Arf1 and Arf3 were highly recruited at the reorganized membranes in the perinuclear area of the MCMV-infected cells already at 6 hpi and even more at 16 hpi. Given that we set up 6 hpi as the earliest time in the development of the pre-AC, which consistently displays the pre-AC structure in ~55% of cells [6,7,12], these data indicate that the accumulation of class I Arfs is associated with the dysregulation of the membranous system that leads to the establishment of the pre-AC. Interestingly, Arf3 was also highly recruited to membranous structures at the cell periphery, including the subplasmalemmal area (Figure 1F), suggesting the dysregulation of the peripheral membranous system outside of the pre-AC of MCMV-infected cells. Note that recruitment of both Arf1 and Arf3 at 6 hpi (Figure 1C,F) was not associated with full cell rounding and contraction, suggesting that the it is a consequence of a membranous organelle rather than cytoskeleton dysregulation.

### 3.2. Increased Expression of Class II Arf Proteins in the Early Phase of Infection and Their Recruitment at Membranes of the Pre-AC

We next analyzed the expression of class II Arfs during the early phase of MCMV infection. Similar to class I Arfs, both of the class II Arf proteins also rapidly increased the expression level (Figure 2). Arf4 increased by 3–4 fold (Figure 2B), whereas Arf5 almost doubled in amount (Figure 2D) at 4 hpi, and both Arfs remained elevated during the entire early phase of infection (Figure 2A,B,D,E).

As demonstrated for the class I Arfs, Arf4, and Arf5 displayed a low level of endogenous expression and membrane recruitment in the MCMV-infected cells immediately after infection (Figure 2C,F), similar to what was observed in the uninfected cells [6]. In contrast to the class I Arfs, Arf4 (Figure 2C) and Arf 5 (Figure 2F) were not highly recruited to the perinuclear membranous organelles of the MCMV-infected cells at 6 hpi, although their expression levels were increased at that time point, espically that of Arf5. At 16 hpi, both Arf4 and Arf5 were highly recruited to membranes in the perinuclear area and were also highly recruited to the peripheral membranous structures of infected cells. These data indicate that the accumulation of class I Arfs coincides with the establishment of the pre-AC structure, whereas class II Arfs are recruited to the pre-AC membranes somewhat later.

### 3.3. Recruitment of Arf6 (Class III Arf Protein) at Membranes of the Pre-AC without Increased Expression in the Early Phase of MCMV Infection

In contrast to the class I and class II Arfs, Arf6 did not show an increased expression level during the early phase of MCMV infection (Figure 3A,B). Although the plasma membrane (PM) is the established localization of Arf6 [35], in uninfected Balb 3T3 cells [6] and MCMV-infected cells immediately after infection (0 hpi), Arf6 was found at membranous structures dispersed through the cell cytoplasm and was not detected on the PM (Figure 3C). At 6 hpi, in most MCMV-infected cells, Arf6 was still at dispersed membranous structures, and was found to be accumulated in the perinuclear area in a small number of cells (Figure 3C). The signal of Arf6 cytoplasmic staining was increased, likely due to the cell contraction and the increasing time of infection coalesced to the juxtanuclear area. At 16 hpi, Arf6 was found accumulated in the juxtanuclear area of MCMV-infected cells. These data indicate that MCMV infection does not change the cellular amount of Arf6 but induces the dysregulation of the Arf6 recruitment cycle, resulting in its accumulation at the membranes of the pre-AC. Since this recruitment is not a prominent feature of 6 h-infected cells, these data suggest that the Arf6 recruitment dysregulation occurs in the sequence after the recruitment of class I Arfs.

Altogether, the upregulation of the cellular levels of class I and class II Arf proteins and the different timings of Arf protein recruitment to the membranes of the pre-AC indicate the dynamic sequence of pre-AC membrane maturation and suggest that Arf proteins may be a crucial host cell factor in pre-AC biogenesis.

### 3.4. All Arf Proteins Accumulate at the Membranes of the Inner Pre-AC, and Arf3 Also Accumulates at the Membranes of the Outer Pre-AC

Considering that all Arf proteins accumulate in the pericentriolar area characteristic for the establishment of the pre-AC at the end of the early phase of MCMV infection (16 hpi), we next examined whether they are located in the outer or inner part of the pre-AC. Namely, our previous study demonstrated that the outer pre-AC is mainly composed of the rearranged Golgi stacks, whereas the inner pre-AC contains many membranous elements derived from EEs, the ERC, and the TGN [6,7,12]. To define Arf protein localization, we performed colocalization analysis of the Arf proteins in MCMV-infected cells that had been infected for 16 h with internalized transferrin receptors (TfRs), the cis-Golgi marker GM130, and the MCMV early protein m06. At 16 hpi, ~60% of internalized TfR is retained in EEs and the rest is retained in the ERC [7], thereby accurately labelling EE- and ERC-derived membranous elements, which, together with TGN-derived membranous structures, build the bulk of the inner pre-AC [6,12]. The GM-130 labels relocated cis-Golgi membranes, which form the inner rim of the outer pre-AC [9,10,11], whereas the m06 protein is retained in the distal Golgi cisternae, mainly in the trans-Golgi [60,61], and thereby is displayed on the outer rim of the outer pre-AC.

At 16 hpi, most of the internalized TfR accumulated in the perinuclear area of MCMV-infected cells (Figure 4A), consistent with our previous results [7,12,58]. The bulk of internalized TfRs compacted around the cell center in the juxtanuclear area. The rest remained in enlarged perinuclear endosomes, as demonstrated in the zoomed area box of the Arf1-stained sample in Figure 4B. The compacted TfRs highly overlapped with all of the Arf proteins, suggesting that all Arf proteins are recruited to the membranous elements of the inner pre-AC (Figure 4B). This resulted in approximately 60% pixel overlap between Arf1, Arf4, Arf5, and Arf6 and internalized TfR (Figure 4C and Appendix A). In addition to the overlap of internalized TfR and Arf3 in the juxtanuclear area (Figure 4B), a significant fraction of Arf3 was displayed outside of this area in the form of a rim that did not show vacuolar appearance. This distribution resulted in the significantly decreased colocalization of Arf3 and internalized TfR to about 40% (Figure 4C and Appendix A). Similar results were observed when the Arf proteins were colocalized with the TGN marker Vti1a (Appendix A). These data suggest that all Arf proteins accumulate at the membranes that form the inner pre-AC and that only Arf3 is substantially recruited at the membranes of the outer pre-AC.

To further examine the localization of Arf proteins in the outer pre-AC, we performed the colocalization of the Arf proteins and the GM130 on the Balb 3T3 cells infected with MCMV at 16 hpi (Figure 5). As expected from colocalization analysis with the TfRs, in MCMV-infected cells, Arf1, -4, -5, and -6 were mainly concentrated within the ring confined by GM130-positive membranes (Figure 5A). The overlaps between these Arfs and GM130 were only observed at the internal rims of GM130-labeled structures, resulting in ~20–30% colocalization being identified by Mander’s coefficient overlap (Figure 5B,C and Appendix A). Given that colocalization was not observed in the outer areas of the GM130-labeled structures, these overlaps are likely the result of a high compacting of internal and outer membranous elements of the pre-AC, which are identified as colocalization during 3D colocalization analysis. In contrast, a significant fraction of Arf3 (~50%) colocalized with GM130 (Figure 5C and Appendix A), although Arf3 also localized in GM130 negative membranous compartments proximally to the cell nucleus and in the membranous structures outside of the GM130-confined ring, including the subplasmalemmal area (Figure 5A). These data suggest that the Arf1, Arf4, Arf5, and Arf6 proteins are mainly recruited to EE-, ERC-, and TGN-derived membranes within the inner pre-AC, whereas Arf3 is primarily recruited to the Golgi-derived membranes of the outer pre-AC. However, Arf3 is also recruited to the inner pre-AC and membranous system at the cell periphery.

To further explore the localization of Arf3, we performed colocalization analysis with the m06 protein, which accumulates at the distal Golgi compartments of MCMV-infected cells [61]. As expected, the minimal overlap between m06 and Arf1, Arf4, Arf5, and Arf6 could be observed (Figure 6A–C and Appendix A). However, the outer pre-AC-associated Arf3 highly colocalized with m06 but not with Arf3 localized within the inner pre-AC and at the cell periphery (Figure 6A,B). These data indicate that Arf3 is highly recruited to the Golgi-derived membranes of the outer pre-AC.

### 3.5. Knockdown of Arf1, Arf3, Arf4, and Arf6 Prevents the Establishment of the Inner Pre-AC

The first event in the formation of pre-AC that can be visualized by immunofluorescence microscopy is an accumulation of expanded EE-ERC-TGN membranes in the juxtanuclear region of the infected cells that represent the formation of the inner part of the pre-AC [6,7,58]. We have shown earlier that Rab10 is the most reliable marker of those earliest events because its signal is barely detectible in uninfected cells. In contrast, in 16 h MCMV-infected cells, Rab10 is accumulated in the juxtanuclear region and displays bright immunofluorescence staining, while other identified markers of pre-AC establishment are already present in that area in uninfected cells or accumulate there significantly in a later stages and therefore do not represent a tool for this study that is as reliable as Rab10 [6]. Therefore, to explore the significance of Arf proteins in the formation of pre-AC, we monitored the accumulation of Rab10 in the pericentriolar region of MCMV-infected cells in which the expression of Arf proteins was silenced by small interfering RNAs (siRNAs). The infection efficiency was monitored by the nuclear expression of the IE1 protein. The specificity and efficiency of siRNA silencing were quantitatively monitored by Western blot analysis in uninfected cells and by the immunofluorescence staining of the Arf proteins in the MCMV-infected cells. The optimized protocols of Arf silencing efficiently depleted Arf proteins 72 h after transfection (Figure 7 and Appendix A), and very few Arf proteins could be detected in transfected cells 16 h after infection with MCMV (Appendix A). These data indicate that the optimized protocols for silencing specifically and efficiently deplete Arf proteins and that the observed upregulation mechanisms induced by MCMV infection cannot override the silencing effect.

As expected, in the control scrambled siRNA-transfected (SCR.) and infected cells immediately after infection with Δm138-MCMV (0 hpi), no cells positive for IE1 were detected, and Rab10 staining showed faint dispersed cytoplasmatic signal without juxtanuclear accumulation (Figure 8). The same was observed in untransfected cells and cells transfected with siRNA targeting Arf proteins (siArf1–6) [62].

At 16 hpi, in cells transfected with SCR. siRNA, 70% of the analyzed cells rounded, accumulated Rab10 in the juxtanuclear region, and expressed bright nuclear staining of IE1 (Figure 8A). Quantification of Rab10 and IE1 expressing cells demonstrated 69 ± 14% of cells with Rab10 juxtanuclear accumulation (Figure 8B and Appendix A) versus 80 ± 12% of IE1 expressing cells (Figure 8C and Appendix A), indicating that the majority of transfected and infected cells developed the inner pre-AC at 16 hpi. This result is similar to that observed in untransfected cells under the same experimental settings, indicating that transfection conditions do not alter the membranous organelle reorganization program in MCMV-infected cells (Appendix A).

Similar to SCR. siRNA-transfected cells, Arf5 knockdown cells (Figure 8A–C) rounded, accumulated Rab10 in the juxtanuclear region (62 ± 7% of cells), and expressed IE1 (84 ± 8% of cells) at 16 hpi. These data indicate that most infected cells undergo the development of the inner pre-AC in the absence of Arf5. In contrast, cells transfected with siArf1, siArf3, siArf4, and siArf6, displayed a lack of cell rounding and significantly reduced the juxtanuclear accumulation of Rab10 (Figure 8A). At 16 hpi, the Rab10 accumulation was observed in 22 ± 12, 30 ± 7, 41 ± 10, and 16 ± 8% of the analyzed cells transfected with siArf1, siArf3, siArf4, and siArf6, respectively, which was significantly lower than the control (SCR.) cells (Figure 8B and Appendix A). However, the ratio of diminished Rab10 accumulation and the expression of IE1 was different among cells with depleted Arf proteins. Although the accumulation of Rab10 was significantly inhibited in siArf3- and siArf4-transfected cells at 16 hpi (Figure 8A,B and Appendix A), the establishment of infection as monitored by the nuclear expression of the IE1 protein was not impaired. Namely, 86 ± 6% of the siArf3 and 85 ± 11% of the siArf4-transfected cells were positive for IE1, similar to that observed in the control cells transfected with scrambled siRNA (SCR.) and siArf5-transfected cells (Figure 8C and Appendix A). However, most of Arf3- and Arf4-depleted cells did not display the cell rounding phenotype (Figure 8A), indicating that Arf3- and Arf4-dependent membranous organelle reorganization within the inner pre-AC are associated with cytopathogenic events related to cytoskeleton rearrangements. In contrast, siArf1 and siArf6-transfected cells (Figure 8A–C and Appendix A), in addition to an almost complete absence of the juxtanuclear accumulation of Rab10, displayed a lack of cell rounding but also a significant decrease in IE1 expression (35 ± 9 and 20 ± 5% positive cells, respectively). These data suggest that both Arf1 and Arf6 are essential for the earlier events in the MCMV replication cycle that occurs during the establishment of infection and before the expression of IE genes.

### 3.6. Knockdown of Arf1 and Arf6 Abolishes the Establishment of MCMV Infection

To further examine the contribution of Arf proteins in the establishment of MCMV infection, cells transfected with control scrambled siRNA (SCR.) or siRNA targeting specific Arf proteins (siArfs) were infected and at 16 hpi and were stained for the expression of the IE1 and m06 proteins. IE1 was expressed within the first hour of infection in the cell nucleus, whereas m06 was also expressed before the initiation of pre-AC formation after 2–3 hpi, but it was localized in the Golgi apparatus that would form the outer part of established pre-AC [60,61]. As expected, the percentage of cells expressing the IE1 and m06 proteins was similar in control cells (SCR.) and Arf3-, Arf4-, and Arf5-depleted cells (Figure 9A,B and Appendix A), indicating normal progression of the immediate–early and early phases of infection in these cells. However, the percentage of IE1 and m06 expressing cells was significantly reduced in Arf1-depleted cells and almost completely abolished in Arf6-depleted cells (Figure 9A,B and Appendix A). This observation was further confirmed by a more sensitive assay: through the infection of cells with the C3X-MCMV (Figure 9C), a recombinant virus generated on the wild-type background by the insertion of a GFP (green fluorescent protein) cassette under the control of the major immediate–early HCMV promoter in front of ie2 gene [55]. After infection with this virus, ~68% of control cells (transfected with scrambled siRNA) were positive for green fluorescence at 16 hpi, as demonstrated by flow cytometric quantification (Figure 9C). In contrast, the green fluorescence signal was only detected in ~37% of siArf1-transfected cells and in ~24% of siArf6-transfected cells. These data confirm our observations from immunofluorescence staining and suggest that Arf1 and Arf6 depletion prevent entry into the immediate–early phase of MCMV gene expression. The fluorescence intensity of GFP expression in siArf1- and siArf6-transfected cells was significantly reduced compared to the fluorescence signal of control cells, indicating that Arf1 and Arf6 depletion significantly reduced the load of the viral genome in the nucleus. This resulted in the drastically reduced expression of IE1 and the almost abolished expression of m06, as demonstrated by Western blot analysis (Figure 10). Altogether, these data suggest that Arf1 and Arf6 are essential for the establishment of infection in the steps before the MCMV genome enters into the nucleus.

## 4. Discussion

This study explored the role of Arf proteins in the biogenesis of pre-AC in MCMV-infected cells. We have shown that already in the early phase of infection, MCMV massively reorganizes the Arf system and increases the levels of class I and class II Arf proteins (Figure 1A,B,D,E and Figure 2A,B,D,F) and leads to accumulation of all Arfs in the perinuclear area of infected cells (Figure 1C,F, Figure 2C,F and Figure 3C). The perinuclear expansion is apparent at 6 hpi, which corresponds with the development of the pre-AC. All Arf proteins are accumulated in the inner area of the pre-AC confined by the outer Golgi stacks and rich in EE/ERC/TGN membranous elements [7,8,9,10,11,12] (Figure 4), indicating that MCMV infection dysregulates Arf recruitment cascades at EE-, ERC-, and TGN-derived membranes. Arf3 is also recruited to membranes of the Golgi in the outer pre-AC (Figure 5 and Figure 6). The recruitment is not synchronous, indicating the sequential dysregulation of the Arf system in the membranous organelle reorganization representing the establishment of the pre-AC. The accumulation of the class I Arfs (Arf1 and Arf3) is evident at 6 hpi, whereas class II and class III Arfs are recruited later. Knockdown experiments, using siRNAs, demonstrated the requirement of Arf1, Arf3, Arf4, and Arf6 for the establishment of pre-AC, as displayed by the accumulation of Rab10 in the juxtanuclear area of the infected cell (Figure 8). However, knockdown of Arf1 and Arf6 affected the earlier stages of infection, prior expression of MCMV immediate early genes (Figure 9 and Figure 10), indicating that they are essential for virus entry or transport to the cell nucleus. Thus, although these Arfs may contribute to the membranous organelle reorganization during pre-AC formation, their direct role in those processes cannot be revealed by knockdown experiments. However, Arf3 and Arf4 knockdown prevented the accumulation of Rab10 in the pericentriolar area of MCMV-infected cells without an evident effect on the establishment of infection and MCMV immediate–early and early gene expression, suggesting their important role in regulating membrane dynamics during pre-AC development.

In uninfected and MCMV-infected Balb 3T3 cells at 0 hpi, immunofluorescence staining of endogenous Arf proteins did not display larger membranous organelles but rather small punctate cytoplasmic structures. This is consistent with their fast turnover at the membranous organelles and aligns with the previously reported data obtained in uninfected cells [63,64]. However, in cells infected with MCMV, the immunofluorescence staining resulted in the bright signal of all of the Arf proteins in the perinuclear area of the cell. The bright staining of the class I Arfs was evident at 6 hpi, whereas class II and III Arfs were displayed later, as demonstrated by staining at 16 hpi. Even though WBs showed significantly increased levels of class I and class II Arfs already at 6 hpi, in contrast to the class I Arfs, at 6 hpi the IF staining of Arf4, and especially that of Arf5, was still weak. This is most likely due to their dispersion in the cytoplasm and the lack of their recruitment and accumulation in the pericentriolar area of MCMV-infected cells at 6 hpi. This conclusion is supported by a significant difference in Arf6 IF staining at 6 hpi and 16 hpi, although Arf6 levels did not significantly change throughout the early phase of MCMV infection.

Although we cannot determine the exact mechanism of Arfs accumulation at the moment, given that the inner pre-AC and AC mainly contain membranous elements derived from the EE, ERC, and TGN [7,9,10,12] and that the outer pre-AC is composed of Golgi stacks, the accumulation of all Arf proteins within the inner pre-AC and the additional recruitment of the Arf3 to the outer pre-AC indicates the massive dysregulation of regulatory loops that control their membrane recruitment. Namely, class I Arfs, Arf1, and Arf3, can be activated at the PM [31,65], EEs [66], the ERC [24,67], and the TGN [17,30]. Arf1 is also associated with the early Golgi [68], whereas Arf3 is associate with the late Golgi [14]. Class II Arfs can be activated at the TGN [69,70], the ERC [25,67], and ER-Golgi intermediate compartment [71]. Arf5 can also be activated at the PM [72]. The class III member Arf6 acts in the feedback axis with Rab35 at the PM [17,65], and the ERC [73,74] and regulates several endocytosis- and exocytosis-related processes within the cell [17,75,76,77]. The regulatory loops that control Arf membrane recruitment are based on the activities of GEFs, cellular proteins that activate the Arfs and facilitate their recruitment to membranes, and GAPs, which dephosphorylate GTP on activated Arfs and thereby initiate their de-recruitment from membranes and their dispersion into the cytosol. Thus, MCMV infection may dysregulate the recruitment of either Arf GEFs or GAPs and thereby cause Arf accumulation at the EE-, ERC-, and TGN-derived membranes within the pre-AC and the accumulation of Arf3 at Golgi membranes that surround the inner pre-AC. Our previous study [6] demonstrated the recruitment of several Arf GEFs within the AC of MCMV-infected cells, suggesting that MCMV infections affect the processes before GEF recruitment. However, the regulatory loops within the Arf system are more complex and not only involve more Arf GEFs and GAPs but also members of the Arl subfamily; thus, an additional systematic study of the Arf regulatory loop recruitment in MCMV-infected cells is required. This study is currently underway in our laboratory.

By recruiting different effector proteins, Arfs control the transport through the entire endosomal system and the TGN [14,17,24,25,67,78,79], whose membrane elements form the inner part of the pre-AC and the AC in the later stages of infection [6,7,9,10]. Arfs are also important for maintaining the structure and function of the Golgi apparatus [11,27,68,71,78,80], whose stacks form the outer pre-AC but retain their basic functions, such as the loading and processing of the MCMV-encoded nonstructural protein m06 and the viral glycoproteins in the late phase of infection [6,7,11]. The recruitment of Arf effectors at membranes depends on the duration of the Arf GTP binding/hydrolysis cycle [14,81]. The accumulation of the Arfs on the membranes of intracellular compartments leads to their expansion and massive accumulation of cargo molecules within them [19,35,41,67]. Therefore, the recruitment of Arf proteins to the inner pre-AC could be one of the reasons for the reorganization and accumulation of the expanded membranous elements of the EE, ERC, and EE-ERC interface in the pericentriolar region of MCMV-infected cells and are therefore important for the biogenesis of pre-AC. Although the activation and recruitment of Arf GTPases on replication organelles have been reported after infection with different viruses [82,83,84,85,86,87,88], their importance in the biogenesis of pre-AC in MCMV-infected cells is understudied.

To address the requirements of Arf proteins in the biogenesis of pre-AC in MCMV-infected cells, we monitored Rab10 accumulation after Arf protein knockdown. We used Rab10 as a readout because of the adverse difference of the immunofluorescence Rab10 signal in uninfected and MCMV-infected Balb 3T3 cells. In uninfected Balb 3T3 cells, the Rab10 signal was barely detectable, which is likely due to a high turnover of Rab10 at membranes, as reported for Madin–Darby Canine Kidney (MDCK) epithelial cells [89]. However, in MCMV-infected cells, Rab10 accumulates in the juxtanuclear area of ~90% of MCMV-infected cells and can be recorded as the earliest and the most reliable event of pre-AC establishment by immunofluorescence studies [6]. The Rab10 accumulation was drastically reduced in Arf1, Arf3, Arf4, or Arf6 knockdown cells, demonstrating the contribution of these Arfs in the biogenesis of the pre-AC. Surprisingly, although Arf5 accumulated at the membranes of the inner pre-AC, it seems that it is not crucial for pre-AC biogenesis. This does not necessarily mean that it is not involved in the formation of pre-AC. It could be that its function is replaceable in Arf5 knockdown Balb 3T3 cells. On the other hand, since in most cases Arf GEFs are not exclusive activators of individual Arfs [20,27], it is also possible that its accumulation is a side effect of the accumulation of Arf GEFs intended for activation of some other Arf protein.

Besides a drastic reduction of Rab10 accumulation, Arf1 and Arf6 knockdowns were associated with decreased immediate–early (IE1) and early (m06) gene expression in MCMV-infected cells, suggesting that Arf1 and Arf6 act at the earliest stages and are essential for the establishment of MCMV infection. These Arfs act at the PM and thus may influence virus attachment to the cell surface, virus entry, and endosomal transport to the cell nucleus or may influence the unpackaging and integrating of the virus genome into the nucleus. Over the years, several PM proteins have been reported to function as CMV receptors with docking or entry-mediating properties, including heparan sulfate proteoglycans, EGFR [90], PDGFR α [91,92,93], integrins [94], Nrp2, CD147 [95,96,97,98], MHC-I molecules [99], CD90 [100], CD147 and CD151 [101]. Thus, the disruption of their endocytic trafficking would alter the virus entry. At the PM, Arf1 and Arf6 regulate both the clathrin-dependent and -independent endocytosis of different cargo molecules, including molecules identified to serve as CMV receptors, such as receptor tyrosine kinases such as PDGFR and EGFR [31,102], integrins [17,90], CD147 [43,81], MHC class I molecules [26,42,103], and GPI-anchored proteins such as CD90 and CD151 [31]. Thus, it is likely that the lack of Arf1 or Arf6 can interrupt the PM-associated processes of MCMV pathogenesis, especially because functions of Arf1 and Arf6 are not reported to be directly linked to the cell nucleus. The role of Arf6 in viral entry was reported for HIV-1 [104,105], Coxsackievirus A9 [106], Vaccinia viruses [107], and Epstein–Barr virus [108] infection, although reported mechanisms might not apply to every infected cell [109].

In cells lacking Arf3 or Arf4, expression of immediate early (IE1) and early (m06) viral proteins was not impaired (Figure 8 and Figure 9), suggesting the regular establishment of MCMV infection. Even more, the fluorescence intensity of GFP expression in siArf3 and siArf4-transfected and C3X-MCMV-infected cells was not significantly changed, compared to the fluorescence signal of the control cells (Appendix A), indicating that Arf3 and Arf4 depletion does not influence the load of the viral genome in the nucleus. However, Arf3 and Arf4 knockdown cells failed to accumulate Rab10 in the juxtanuclear region (Figure 8 and Figure 9) and to establish pre-AC. Although the exact mechanisms by which Arf3 and Arf4 affect the biogenesis of pre-AC have yet to be elucidated, one of the possible mechanisms could be related to the organization of the microtubular network. A recent report demonstrates that cytoskeletal rearrangement and destabilization of microtubules may be an essential step in the pre-AC biogenesis [110]. Since only slight cell rounding was observed in Arf3 and Arf4 knockdown cells (Figure 8 and Figure 9), it is possible that the activity of Arf3 and Arf4 is associated with processes that influence the destabilization of the microtubular network in MCMV-infected cells during the formation of the pre-AC and that Arf3 and Arf4 knockdowns prevent the destabilization of the microtubules, which, in turn, blocks the formation of the pre-AC. Another possibility is an indirect contribution of Arf4 in pre-AC biogenesis. In uninfected cells, activated Arf1 is mainly found on the Golgi membranes, and its function is related to the linking and maintaining of the structure of the Golgi [111,112,113], while it is not essential for the *cis*-Golgi association of GM130 [114]. However, it was also shown that activated Arf4 facilitates the recruitment of the Arf1 GEFs Big1 and Big2 [27,115,116] to the TGN and consequently increases the activation of Arf1 on the TGN membranes [70]. Thus, the recruitment of Arf4 on the inner pre-AC membranes during MCMV infection could be a reason for the increased recruitment of Arf1 to the TGN and the reduced activation of Arf1 on the Golgi membranes that could have an influence on Golgi unlinking, one of the steps of pre-AC formation [6,7,11,12]. Thus, in cells lacking Arf4, the activation of Arf1 on the Golgi membranes would be less reduced, which could impair Golgi unlinking and pre-AC establishment.

In conclusion, our study demonstrates a significant role of Arf1, Arf3, Arf4, and Arf6 in the pathogenesis of CMV infection and host cell reorganization during the early phase of infection. Our study identifies at least Arf3 and Arf4 as required factors in the membranous organelle rearrangement during the establishment of the pre-AC. Arf1 and Arf6 may be also important players in this process. However, they substantially contribute to the earlier processes, and therefore, their direct role in the reorganization of the membrane dynamics associated with pre-AC formation cannot be demonstrated by knockdown experiments, which is an obstacle often associated with research on the role of host cell factors during viral infection.

## 5. Conclusions

Our study demonstrated increased levels of class I and II Arf proteins in the early phase of MCMV infection, the accumulation of all Arfs at the membranes of the pre-AC, and the requirement of Arf1, Arf3, Arf4, and Arf6 for the development of the MCMV pre-AC. However, our study also revealed that the contribution of Arf1 and Arf6 is most likely associated with the earliest stages of infection, either in terms of virion entry or the establishment of infection. Given that many processes of the AC biogenesis are conserved among beta-herpesviruses, our data may contribute to understanding the biogenesis of the HCMV assembly compartment and identifying host cell processes that may be targeted for the development of antivirals.

## Figures and Tables

**Figure 1 life-11-00867-f001:**
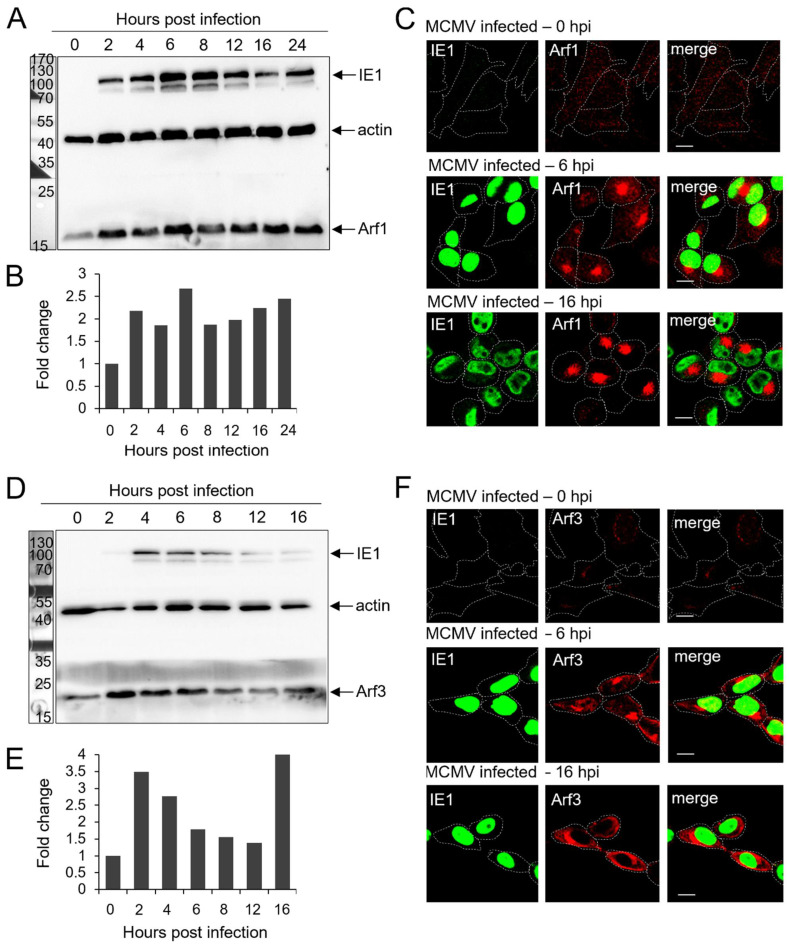
Expression levels of class I Arf proteins and their intracellular localization in the MCMV-infected cells during the early phase of infection. Balb 3T3 cells were infected with Δm138-MCMV, and the expression level and intracellular recruitment of Arf1 and Arf3 in the early phase of MCMV infection (0–16 hpi) were analyzed by Western blot and immunofluorescence. Representative Western blots of Arf1 (**A**) and Arf3 (**D**) expression in the early phase of MCMV infection are presented. Actin was used as a loading control, and IE1 expression was used as an infection control. Protein markers are shown on the left. Quantitative analysis of Arf1 (**B**) and Arf3 (**E**) expression levels was performed as described in *Materials and Methods.* Results are expressed as a fold change relative to Arf expression in 0 hpi MCMV-infected cells. Immunofluorescence images of Arf1 (**C**) and Arf3 (**F**) expression relative to the nuclear staining of the immediate–early 1 (IE1) MCMV protein during the early phase of infection. Δm138-MCMV-infected Balb 3T3 cells were fixed at 0, 6, and 16 hpi, permeabilized, and stained against IE1 and either Arf1 or Arf3. The antibodies were visualized with the appropriate fluorochrome-conjugated non-crossreactive secondary reagents. Cell borders are indicated by dashed lines. Bars, 10 μm.

**Figure 2 life-11-00867-f002:**
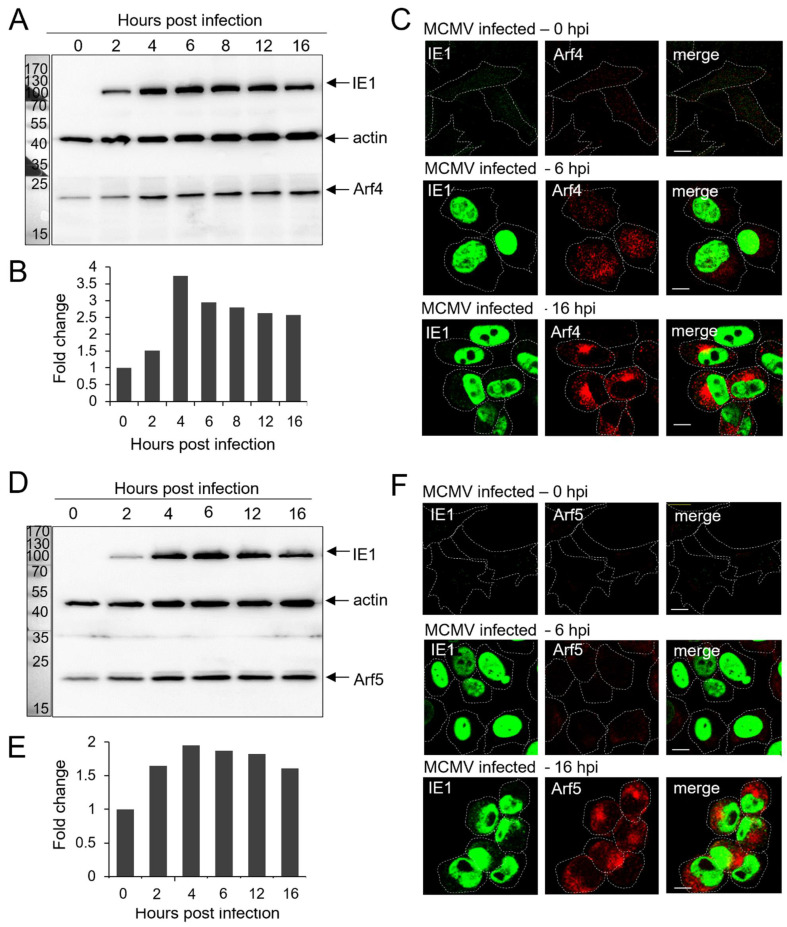
Expression levels of class II Arf proteins and their intracellular localization in the MCMV-infected cells during the early phase of infection. Balb 3T3 cells were infected with Δm138-MCMV, and the expression level and intracellular recruitment of Arf4 and Arf5 in the early phase of MCMV infection (0–16 hpi) were analyzed by Western blot and immunofluorescence. Representative Western blots of Arf4 (**A**) and Arf5 (**D**) expression in the early phase of MCMV infection are presented. Actin was used as a loading control, and IE1 expression was used as a control of infection. Protein markers are shown on the left. Quantitative analysis of Arf4 (**B**) and Arf5 (**E**) expression levels was performed as described in *Materials and Methods.* Results are expressed as a fold change relative to Arf expression in 0 hpi MCMV-infected cells. Immunofluorescence images of Arf4 (**C**) and Arf5 (**F**) expression relative to the nuclear staining of immediate–early 1 (IE1) MCMV proteins during the early phase of infection. Δm138-MCMV-infected Balb 3T3 cells were fixed at 0, 6, and 16 hpi, permeabilized, and stained against IE1 and either Arf4 or Arf5. The antibodies were visualized with the appropriate fluorochrome-conjugated non-crossreactive secondary reagents. Cell borders are indicated by dashed lines. Bars, 10 μm.

**Figure 3 life-11-00867-f003:**
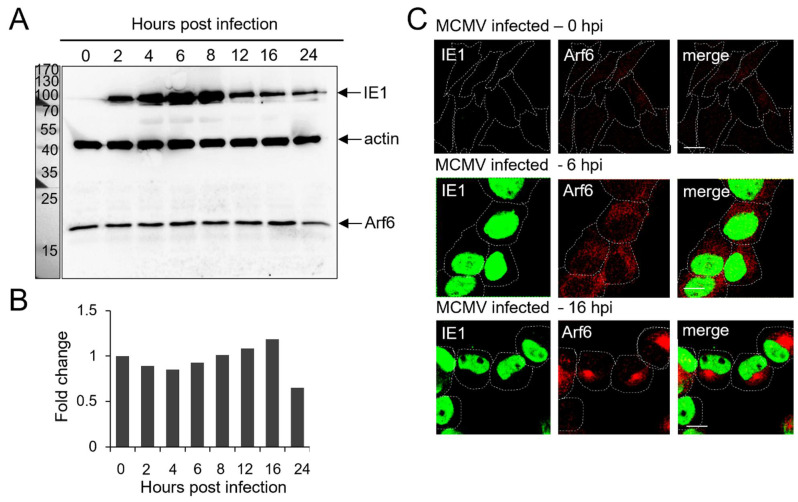
Expression levels of the class III Arf protein and its intracellular localization in the MCMV-infected cells during the early phase of infection. Balb 3T3 cells were infected with Δm138-MCMV, and the expression level and intracellular recruitment of Arf6 in the early phase of MCMV infection (0–16 hpi) were analyzed by Western blot and immunofluorescence. Representative Western blots of Arf6 (**A**) expression in the early phase of MCMV infection are presented. Actin was used as a loading control, and IE1 expression was used as an infection control. Protein markers are shown on the left. (**B**) Quantitative analysis of Arf6 expression levels was performed as described in *Materials and Methods.* Results are expressed as a fold change relative to Arf6 expression in 0 hpi MCMV-infected cells. Immunofluorescence images of Arf6 (**C**) expression relative to the nuclear staining of immediate–early 1 (IE1) MCMV proteins during the early phase of infection. Δm138-MCMV-infected Balb 3T3 cells were fixed at 0, 6, and 16 hpi, permeabilized, and stained against IE1 and Arf6. The antibodies were visualized with the appropriate fluorochrome-conjugated non-crossreactive secondary reagents. Cell borders are indicated by dashed lines. Bars, 10 μm.

**Figure 4 life-11-00867-f004:**
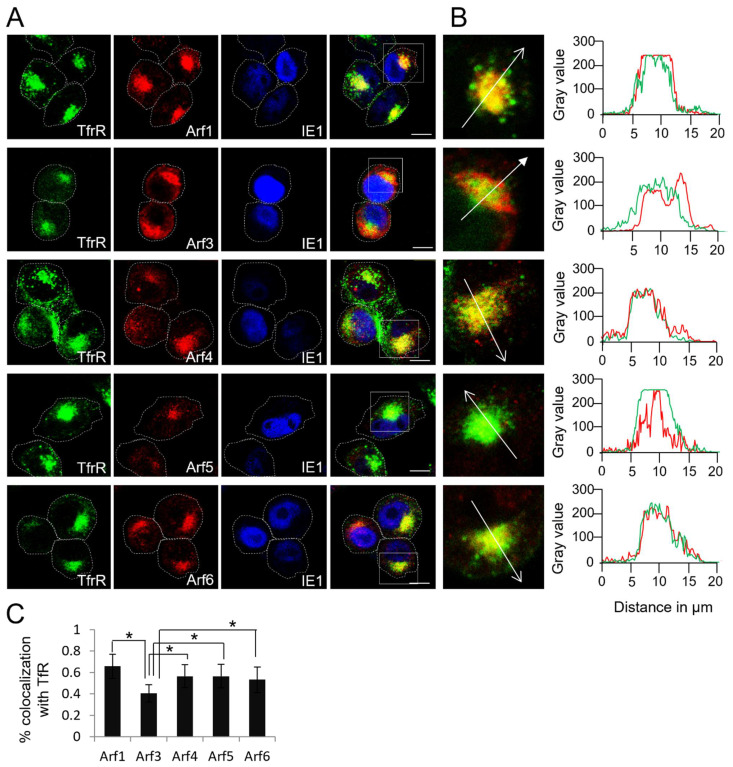
Arf proteins enriched within the inner part of the pre-AC. (**A**) At 15 hpi, Δm138-MCMV-infected Balb 3T3 cells were incubated 45 min with Tf-AF^488^ at 37 °C (green fluorescence), fixed, permeabilized, and stained against Arf proteins (red fluorescence) and IE1 (blue fluorescence). Triple-stained images are shown (focal plane across the mid-section of the cells). Fine dashed lines indicate cell borders. Full-lined boxes indicate the zoomed area. Bars, 10 μm. (**B**) Zoomed images were analyzed by plotting fluorescence intensity profiles along white arrow lines on the MaxEntropy threshold of the images. (**C**) Images were analyzed through the entire z-stack for colocalization using Mander’s coefficients of pixel overlap. Data represent mean ± STDEV per cell (*n* = 10–15). Asterisks indicate statistical significance (*: *p* < 0.005).

**Figure 5 life-11-00867-f005:**
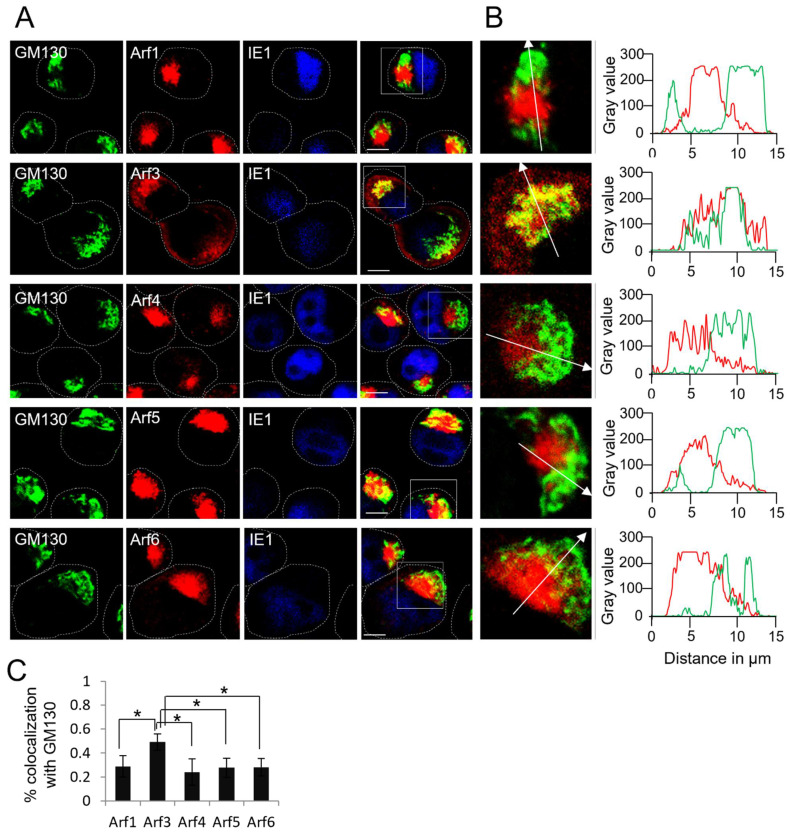
Arf proteins, except Arf3, are mainly excluded from the GM130-positive membraneous compartments of the pre-AC. (**A**) Δm138-MCMV-infected Balb 3T3 cells were fixed at 16 hpi, permeabilized, and stained against Arf protein (red fluorescence), GM130 (green fluorescence), and IE1 (blue fluorescence). Focal planes across the mid-section of the cells are shown. Dashed lines indicate cell borders. Full-lined boxes indicate the zoomed area. Bars, 10 μm. (**B**) Zoomed images were analyzed by plotting fluorescence intensity profiles along white arrow lines on the MaxEntropy image threshold. (**C**) Images were analyzed through the entire z-stack for colocalization using Mander’s coefficients of pixel overlap. Data represent mean ± STDEV per cell (*n* = 10–14). Asterisks indicate statistical significance (*: *p* < 0.005).

**Figure 6 life-11-00867-f006:**
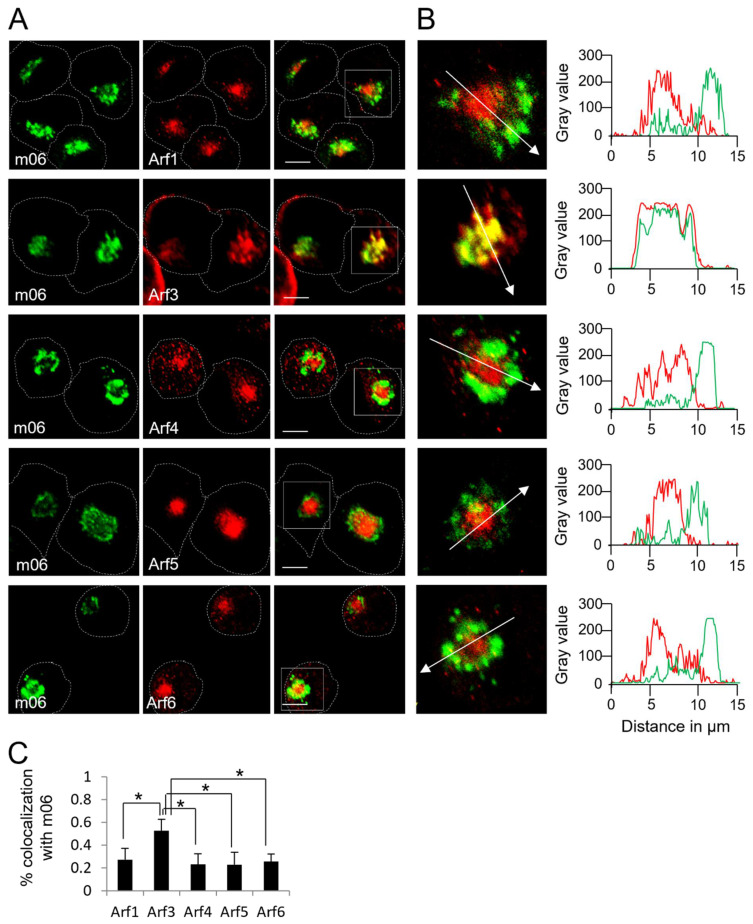
Arf3 protein is recruited to membranes of the trans-Golgi elements of the outer pre-AC. (**A**) Δm138-MCMV-infected Balb 3T3 cells were fixed at 16 hpi, permeabilized, and stained against Arf protein (red fluorescence) and m06 (green fluorescence). Focal-plane images across the mid-section of the cells are shown. Dashed lines indicate cell borders. Full-lined boxes indicate the zoomed area. Bars, 10 μm. (**B**) Zoomed images were analyzed by plotting fluorescence intensity profiles along white arrow lines on the MaxEntropy threshold of images. (**C**) Images were analyzed through the entire z-stack for colocalization using Mander’s coefficients of pixel overlap. Data represent mean ± STDEV per cell (*n* = 10–13). Asterisks indicate statistical significance (*: *p* < 0.005).

**Figure 7 life-11-00867-f007:**
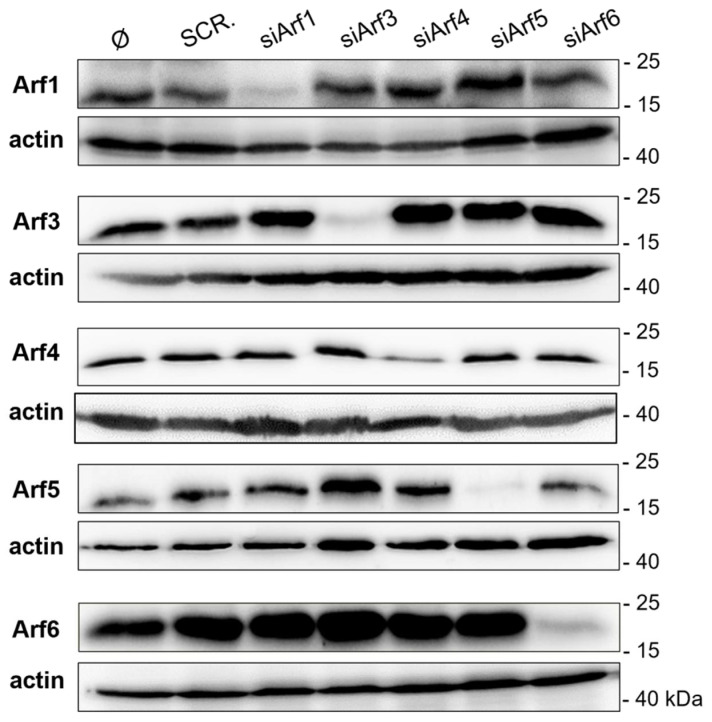
Efficiency and specificity of Arf1, Arf3, Arf4, Arf5, and Arf6 silencing. Balb 3T3 cells were transfected with negative control scrambled siRNA (SCR.) or siRNAs targeting Arf proteins (siArf1–6). After 72 h, cell lysates were analyzed by Western blot and stained with specific antibodies against Arf proteins. Actin was used as a loading control.

**Figure 8 life-11-00867-f008:**
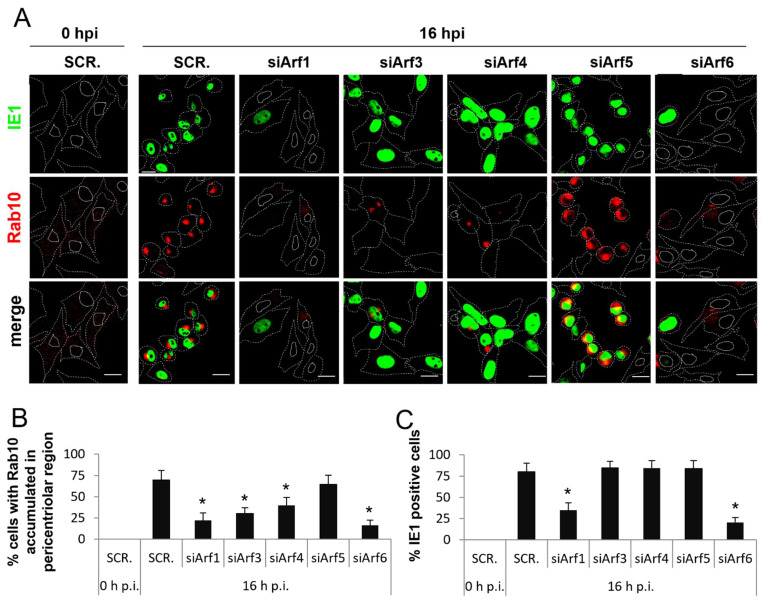
Knockdown of Arf1, Arf3, Arf4, and Arf6 prevents the development of the pre-AC. **(A)** Balb 3T3 cells were transfected with negative control scrambled siRNA (SCR.) or siRNAs targeting Arf1–6 (siArf1–6) and after 72 h infected with Δm138-MCMV. At 0 and 16 hpi, cells were fixed, permeabilized, and stained against Rab10 (red fluorescence) and IE1 (green fluorescence). Cell borders are indicated by dashed and nuclei by full lines. Representative images with focal planes across the mid-section of the cells are shown. Bars, 20 μm. Percentages of cells with the juxtanuclear Rab10 accumulation (**B**) and nuclear IE1 expressions (**C**) were determined as described in *Materials and Methods,* and data are presented as the mean ± STDEV. Asterisks above error bars indicate statistical significance (*: *p* < 0.005) when compared to the control (SCR.).

**Figure 9 life-11-00867-f009:**
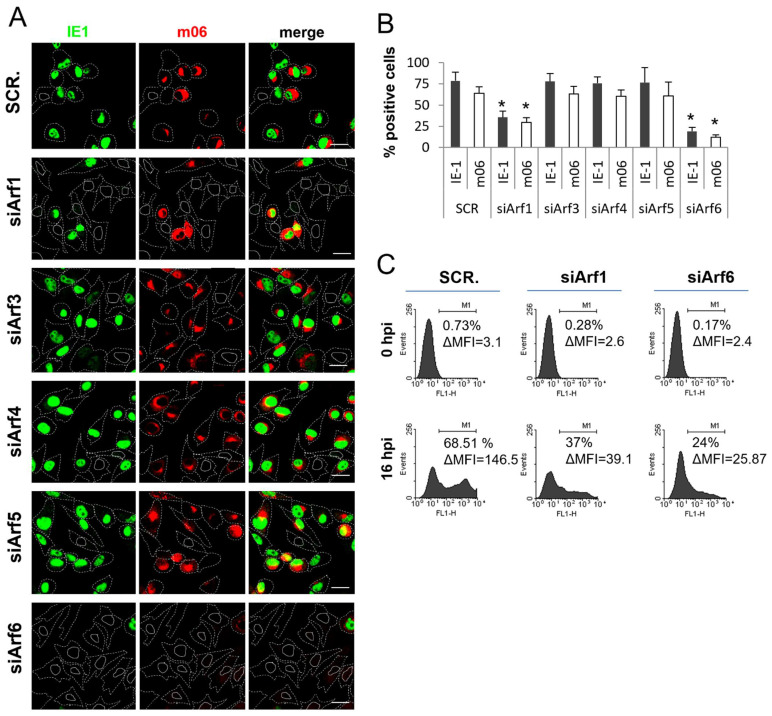
Knockdown of Arf1 and Arf6 inhibits the establishment of MCMV infection. (**A**) Balb3T3 cells were transfected with negative control scrambled siRNA (SCR.) or siRNAs targeting Arf1–6(siArf1–6) and after 72 h infected with Δm138-MCMV. At 16 hpi, cells were fixed, permeabilized, and stained against m06 (red fluorescence) and IE1 (green fluorescence). Cell borders are indicated by dashed lines and nuclei by full lines. Representative images with focal planes across the mid-section of the cells are shown. Bars, 20 μm. (**B**) The percentage of IE1 and m06 positive cells was determined as described in *Materials and Methods.* Data represent mean ± STDEV, and asterisks above error bars indicate statistical significance (*: *p* < 0.005) when compared to the control (SCR.). (**C**) Balb 3T3 cells transfected with scrambled siRNA (SCR.), siArf1, or siArf6 were infected with C3X-GFP MCMV 72 h after transfection, and at 0 and 16 hpi, the GFP fluorescent signal was analyzed by flow cytometry. A representative of three independent experiments is shown.

**Figure 10 life-11-00867-f010:**
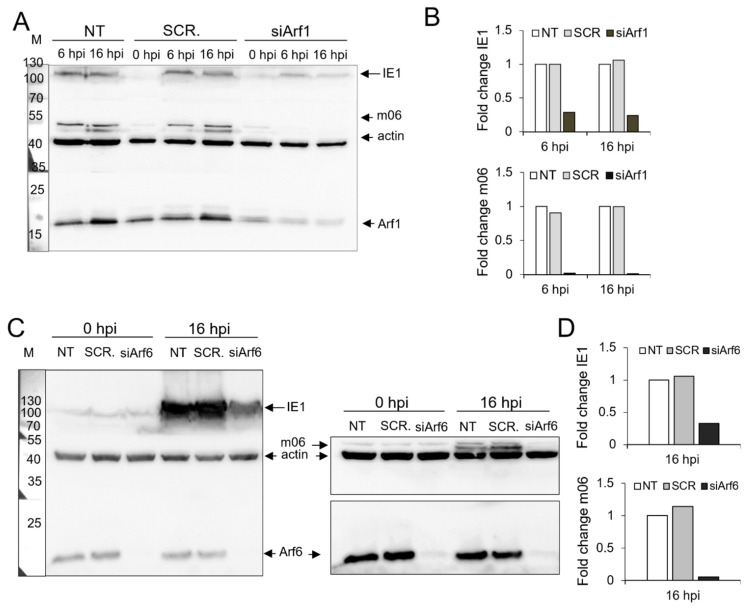
Arf1 and Arf6 are essential for the establishment of MCMV infection. (**A**) Untransfected (NT)**,** scrambled siRNA- (SCR.), or siRNA targeting Arf1- (siArf1) transfected Balb 3T3 cells were infected with Δm138-MCMV 72 h after transfection. At 0, 6, and 16 hpi, the expression of IE1 and m06 was analyzed by Western blot. Actin was used as a loading control, and Arf1 expression was used as a silencing efficiency control. A representative experiment is shown. (**B**) Quantitative analysis of IE1 and m06 expression levels in NT, SCR.-, and siArf1-transfected cells at 0, 6, and 16 hpi was performed as described in *Materials and Methods.* Results are expressed as a fold change relative to IE1 or m06 expression in NT cells at the same time of infection. (**C**) NT, SCR.-, or siArf6-transfected Balb 3T3 cells were infected with Δm138-MCMV 72 h after transfection, and at 0 and 16 hpi, the expression of IE1 and m06 was analyzed by Western blot. Actin was used as a loading control, and Arf6 expression was used as a silencing effiency control. The right panel represents WB signals after additional staining with the anti-m06 antibody and at longer exposures. A representative experiment is shown. (**D**) Quantitative analysis of IE1 and m06 expression levels in NT, SCR-., and siArf6-transfected cells at 0 and 16 hpi was performed as described in *Materials and Methods.* Results are expressed as a fold change relative to IE1 or m06 expression in NT cells at the same time of infection.

## Data Availability

The data presented in this study are available upon request from the corresponding author.

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
