# Peer review of "Arf GTPases Are Required for the Establishment of the Pre-Assembly Compartment in the Early Phase of Cytomegalovirus Infection"

_life, 2021, doi:10.3390/life11080867_

Round 1
Reviewer 1 Report
This manuscript by Pavisic, et al. investigates the role of Arf GTPases in establishing the pre-assembly compartment during cytomegalovirus infection. The authors show that Arf proteins are overexpressed and recruited to the perinuclear pre-AC early on in infection and that knockdown of several of these Arf proteins block pre-AC formation or establishment of infection. I believe that this manuscript could be improved by addressing the following points:
- The authors suggest that upregulation of Arf proteins during MCMV infection is the result of disrupted degradation of the endogenous proteins. While this seems likely, it would be best to show this using cyclohexamide chase experiment, etc.
- Are these overexpressed proteins the active or inactive form? Can this be teased out somehow? Identifying this would greatly add to this story. Similarly, do you know how they are upregulated? Is there a specific viral protein that is important for this process?
- It would be helpful if the results of Figures 1-3 referring to the percentage of cells showing over recruitment of Arf proteins to the pre-AC was quantified. For example, the authors say “in a small number of cells…” on line 310. Please be more specific.
- Is there a reason that the authors did not test Arf2 along with the others?
- Do the authors have a better example of Arf4 knockdown than the one shown in Figure 7? This does not look very convincing and may leave room for readers to suggest off-target affects are involved in the observed phenotypes.
- It is interesting that Arf5 is overexpressed and recruited to the pre-AC, but it’s knockdown does not affect the development of the pre-AC like the others. Could you please add some discussion on why you think this one behaves differently?
- Figure 10 seems to be either mislabeled or just confusing. It should be described in more detail in the results section where each panel is at least discussed. Why does the untransfected sample have a 3hpi time point, but the others don’t? Also, the blot does not show a 16hpi time point for the scrambled sample. This is an important control if you are going to report on the results of Arf1 at this time. Lastly, it appears that the scrambled control blocks expression of m06 at 6hpi compared to the NT sample in Figure 1A, which should not be the case. I suggest replacing this blot all together with one showing scrambled at 16hpi and no effect on m06.
- Minor- Several words in section 3.1 are italicized. Is this on purpose?
Author Response
- The authors suggest that upregulation of Arf proteins during MCMV infection is the result of disrupted degradation of the endogenous proteins. While this seems likely, it would be best to show this using cyclohexamide chase experiment, etc.
RESPONSE: Thank you for your suggestion. We already tried to use cycloheximide chase experiments, but unfortunately, cycloheximide blocks the synthesis of viral proteins and therefore prevents the establishment of CMV infection. Therefore, effects of CMV on Arf proteins expression levels cannot be analyzed in cells treated with cycloheximide or other inhibitors of protein synthesis. Thus, our conclusions are based on the transcriptome analysis of CMV infected cells that shown that CMV infection does not significantly change mRNA levels of none of Arf proteins in the early phase of infection (REF: 6), suggesting that increased protein levels of Arf1, Arf3, Arf4, and Arf5 are the consequence of inhibited degradation and not altered synthesis. We changed this part in the description of the results (lines 241-248).
- Are these overexpressed proteins the active or inactive form? Can this be teased out somehow? Identifying this would greatly add to this story. Similarly, do you know how they are upregulated? Is there a specific viral protein that is important for this process?
RESPONSE: Thank you for your question. Determining the degree of activation as well as the mechanism of activation/inactivation of Arf proteins during CMV infection is a complex issue on which we are currently working on in our laboratory. One way to determine if Arf proteins are activated is to determine if Arf GEF proteins (the proteins that mediate their activation) and effector proteins are present at the site of Arf protein accumulation during the MCMV infection. Arf effectors will accumulate at the same site as Arfs only if the Arf proteins are in the active form. We have previously shown that major Arf GEF proteins (e.g. Big1, Big2, GEP-100, cytohesin, etc.) that mediate the activation of Arf proteins are also accumulated in the pericentriolar region of MCMV infected cells, as well as effector molecules of Arf proteins, such as Epi64, Rab8, Rab36. Those results suggest that Arf proteins accumulated in the pericentriolar region of MCMV infected cells are in the activated form, but we are currently working on the establishment of the methods by which we will be able to determine the degree of Arfs activation more precisely during MCMVC infection. We added this part in the discussion (610-635). At the moment, we do not have data that could elucidate the exact mechanism of their activation. One of the possible mechanisms is certainly their activation via one of the viral proteins who could have Arf GEF activity, like it is shown for HIV-1 Nef.
- It would be helpful if the results of Figures 1-3 referring to the percentage of cells showing over recruitment of Arf proteins to the pre-AC was quantified. For example, the authors say “in a small number of cells…” on line 310. Please be more specific.
RESPONSE: Thank you for your suggestion. We specified the percentage of cells showing over recruitment of Arf proteins in the pre-AC in MCMV infected cells (lines 228 and 273)
- Is there a reason that the authors did not test Arf2 along with the others?
RESPONSE: We did not test Arf2 because murine cells do not express Arf2. We added this information in the introduction (lines 73 and 74).
- Do the authors have a better example of Arf4 knockdown than the one shown in Figure 7? This does not look very convincing and may leave room for readers to suggest off-target affects are involved in the observed phenotypes.
RESPONSE: Thank you for your comment. We changed the WB showing Arf4 knockdown in Figure 7, and 3 different WBs of Arf4 silencing can be found in Supplementary figure 3. Although in some WBs Arf4 signal was still detectible (~15%), when we analyzed Arf4 silencing efficiency by IF, Arf4 was not detectible in more than 90% of Arf4 knockdown cells, even the MCMV infected ones (Supplementary figure 5). Similar efficiency of Arf4 silencing obtained at the same time by WB and IF can be seen in previously published papers (e.g. Kudelko M et al., JBC, 2012). Regardless the small amount of Arf4 was still detectible by WB in Arf4 knockdown cells, we believe that phenotype observed in Arf4 knockdown, as well as phenotypes observed in cells lacking other Arf proteins, are not the consequence of off-targeting effects since silencing of either of the Arf proteins did not decrease protein levels of other Arf proteins (Figure 7) although six mammalian Arfs are highly conserved, sharing > 65% sequence identity and Arf4 and Arf5 share over 90% identity at the amino acid level (Follit JA et al., PLOS Genetics, 2014).
- It is interesting that Arf5 is overexpressed and recruited to the pre-AC, but it’s knockdown does not affect the development of the pre-AC like the others. Could you please add some discussion on why you think this one behaves differently?
RESPONSE: Thank you for your suggestion. We added this part in the discussion (lines 748-754)
- Figure 10 seems to be either mislabeled or just confusing. It should be described in more detail in the results section where each panel is at least discussed. Why does the untransfected sample have a 3hpi time point, but the others don’t? Also, the blot does not show a 16hpi time point for the scrambled sample. This is an important control if you are going to report on the results of Arf1 at this time. Lastly, it appears that the scrambled control blocks expression of m06 at 6hpi compared to the NT sample in Figure 1A, which should not be the case. I suggest replacing this blot all together with one showing scrambled at 16hpi and no effect on m06.
RESPONSE: Thank you for your suggestion. The WB shown in Figure 10A was not mislabeled. Simply, we did not have enough space on one gel to put all the samples that we wanted to analyze and the effects of control SCR. siRNA at 16 hpi are also shown in Figure 10C (right panel). Although it might seem that at 6 hpi, the expression of m06 is lower in cells transfected with SCR. siRNA then in nontransfected (NT) cells, quantification of expression relative to actin (previous Figure 10B, now supplementary figure 10B) shows that m06 expression is not significantly altered with SCR. siRNA. However, we agree that the way the data were presented can be confusing. Therefore, another WB that shows 6 and 16 hpi for NT, SCR., and Arf1 siRNA transfected cells is now shown in Figure 10A, as well as its quantification in Figure 10B, and previous WB is now shown in Supplementary figure 10 so as not to lead to confusion.
- Minor- Several words in section 3.1 are italicized. Is this on purpose?
RESPONSE: Thank you for noticing. No, this was no on purpose. We corrected it.
Reviewer 2 Report
This manuscript is a comprehensive survey of the expression and localization of Arf family members during early events in MCMV infection. The authors show that all Arf family members are recruited to the assembly compartment and that all except Arf6 are increased during infection. Further, they carefully examine the complex organization of the assembly compartment and find differential localization on the endomembranes. For the most part, the data is very high quality. A couple of revisions are suggested.
Major points
- For much of the biochemical data, there is no statistical analysis of the quantification. This should be corrected.
- The discussion is primarily a restatement of the results. There is no integration into the CMV literature or discussion of how these factors might impact CMV assembly.
Minor points
- It is stated in the discussion that MCMV infection "induces overexpression" of the Arf proteins. This is an overstatement. It could be decreased degradation or increased activation. This should be clarified.
- "Over recruited" is used throughout the manuscript, but is not a useful descriptor. Recruited, accumulates, concentrated etc might be used in place.
Author Response
Major points
- For much of the biochemical data, there is no statistical analysis of the quantification. This should be corrected.
RESPONSE: Thank you for your comment. We added statistical analysis of quantifications in Figures 3, 4, and 5. Also, we added supplementary figure 8 that shows statistical analysis of WB results of Arf protein expression during CMV infection that is related to figures 1, 2, and 3.
- The discussion is primarily a restatement of the results. There is no integration into the CMV literature or discussion of how these factors might impact CMV assembly.
RESPONSE: Thank you for your suggestion. We modified the discussion and tried to better explain the impact of Arfs on pre-AC development and consequently on the CMV assembly.
Minor points
- It is stated in the discussion that MCMV infection "induces overexpression" of the Arf proteins. This is an overstatement. It could be decreased degradation or increased activation. This should be clarified.
RESPONSE: Thank you for your suggestion. We tried to explain it better (lines 241-248).
- "Over recruited" is used throughout the manuscript, but is not a useful descriptor. Recruited, accumulates, concentrated etc might be used in place.
RESPONSE: Thank you for your suggestion. We replaced over recruited either with recruited or accumulated.
Reviewer 3 Report
In the manuscript entitled "Arf GTPases are required for the establishment of the pre-assembly compartment in the early phase of cytomegalovirus infection" Pavisic and colleagues show that the Arf GTPases co-localize with membranes of the AC and that knockdown of the Arfs have differential effects on MCMV infection. The findings are of interest and are backed by quality data.
Major Point
Knockdown of Arf3 and Arf4 have a significant effect on Rab10 localization to the pericentriolar region (Figure 8B). What is the effect on other proteins, particularly viral proteins? Do they localize properly in the Arf3 and Arf4 knockdowns? For example, Figure 9A seems to show m06 in a pericentriolar region in many of the Arf3 and Arf4 knockdown cells.
While the impact of Arf1 and Arf6 on establishing infection is clear, what is the effect of of Arf3 and Arf4 knockdown on viral titers. Linking the dysregulation of membrane trafficking to infection outcome is important to justify the conclusion that Arf3 and Arf4 play an important role in the pathogenesis of CMV infection (lines 679-680).
Minor Points
Use of the trendlines seem unnecessary and they are actually misleading in some cases as they do not follow the data. For example in Figure 1E, the trendline is going in the opposite direction as the data due to the marked increase at 16 hpi. As they don't add meaning to the data (the reader can see and interpret the data without them), they should be removed but if the trendlines are used, at the minimum an R-squared value should be included (which seems like it would be very low in several cases).
There appears to be a discrepancy between Figures 2D & 2F. While the western shows equivalent levels of Arf5 at 6 and 16 hours post infection, there is a drastic difference in the IF staining. The authors should address this discrepancy.
The legend in Figure 9 references a western in (D), which does not exist.
Author Response
Major Point
- Knockdown of Arf3 and Arf4 have a significant effect on Rab10 localization to the pericentriolar region (Figure 8B). What is the effect on other proteins, particularly viral proteins? Do they localize properly in the Arf3 and Arf4 knockdowns? For example, Figure 9A seems to show m06 in a pericentriolar region in many of the Arf3 and Arf4 knockdown cells.
RESPONSE: Thank you for your comment. In this paper, we analyzed Arf proteins in the early phase of MCMV infection (up to 16 hpi) when only immediate-early and early viral proteins are expressed. At this time point, viral proteins anylyzed localize in the nucleus of infected cells. The exception is m06 which even before the initiation of pre-AC formation (at 2-3 hpi) localize to the pericentriolar region. m06 localizes in the Golgi apparatus which leter forms the outer part of established pre-AC and retains its basic function, such as loading and processing of MCMV-encoded nonstructural protein m06 and viral glycoproteins in the late phase of infection [REF: 6,7,11]. We added those information in lines 509-512 and 776-778. Therefore, expression and localization of immediate early and early viral proteins do not provide information about the establishment of pre-AC, and we followed the expression of IE1 (as a representative of immediate early viral proteins) and m06 (as a representative of early phase viral proteins) in order to follow the establishment of MCMV infection in Arf knockdown cells.
We have shown earlier that Rab10 is the most reliable marker of the earliest steps of the establishment of the MCMV pre-AC (Lučin P et al., ) and therefore, we used it to analyze establishment pre-AC in the early phase of MCMV infection in Arf knockdown cells. Rab10 does not stain distinct membranous entities in untreated Balb 3T3 cells, presumably because it is recruited to short-lived membranous intermediates (Babbey et al., 2006). Thus, reorganization events associated with the recruitment of Rab10 in the pericentriolar region of infected cells are clearly distinguishable and can be identified as the earliest events by immunofluorescence. Many other cellular proteins also accumulate in the pericentriolar area of the infected cell, especially proteins that characterize membranes of the EE-ERC-TGN interface, including Arf proteins and their GEFs (see Lučin et al., 2020). Thus, Arf proteins and their GEFs can also be used as reliable markers for the earlies identification of the pericentriolar accumulation events. However, these proteins are the subject of the present study and were not used as markers. Other proteins, exemplified by Rab11, EEA1, or Rab8, are also accumulated. However, they are already membrane-recruited in the juxtanuclear area of the uninfected cell and cannot be used as markers for the identification of the earliest events in immunofluorescence studies. We tried to explain this in lines 434-441 and 738-746. Accumulation of Rab36 is as good marker as Rab10 regarding the contrast to an uninfected cell. However, it appears that Rab36 accumulation is lagging behind Rab10 and cannot be identified in as large a number of cells as Rab10 in the earliest times of pre-AC establishment. Similar is also to Evectin-2. Both of these two markers are very useful but at later times. We also tested the effects of Arf3 and Arf4 knockdown on the accumulation of Rab36 in the pericentriolar region of MCMV infected cells. Both knockdowns inhibited the accumulation of Rab36 (unpublished observations), which goes in line with results obtained with Rab10.
In this paper we show that both, immediate early (IE1) and early (m06) phase MCMV viral proteins are normally expressed in Arf3 and Arf4 knockdown cells (Figures 8, 9, supplementary figure 5), and at the same time there is no accumulation of Rab10 in the pericentriolar region on MCMV infected cells (Figure 8). These results suggest that Arf3 and Arf4 are not crucial for establishment of MCMV infection but are required for establishment of pre-AC. We certainly are planning to analyze the effects of Arf3 and Arf4 knockdown on the expression of late phase MCMV viral proteins that are structural proteins and accumulate in the pericentriolar region in the area of AC formation in the late phase of MCMV infection. It will also be important to see if the production of infective virions is enabled in Arf3 and Arf4 knockdown cells.
- While the impact of Arf1 and Arf6 on establishing infection is clear, what is the effect of Arf3 and Arf4 knockdown on viral titers. Linking the dysregulation of membrane trafficking to infection outcome is important to justify the conclusion that Arf3 and Arf4 play an important role in the pathogenesis of CMV infection (lines 679-680).
RESPONSE: Thank you for your comment. Viral titers are not significantly altered in Arf3 and Arf4 knockdown cells. We added a new supplementary figure 9 showing those results. We also discussed this in lines 778-782.
Minor Points
- Use of the trendlines seem unnecessary and they are actually misleading in some cases as they do not follow the data. For example in Figure 1E, the trendline is going in the opposite direction as the data due to the marked increase at 16 hpi. As they don't add meaning to the data (the reader can see and interpret the data without them), they should be removed but if the trendlines are used, at the minimum an R-squared value should be included (which seems like it would be very low in several cases).
RESPONSE: Thank you for your suggestion. We removed trend lines from Figures 1B, 1E, 2B, 2E and 3B.
- There appears to be a discrepancy between Figures 2D & 2F. While the western shows equivalent levels of Arf5 at 6 and 16 hours post infection, there is a drastic difference in the IF staining. The authors should address this discrepancy.
RESPONSE: Thank you for your suggestion. We emphasized this observation in lines 293-294 and addressed this in discussion (lines 603-609).
- The legend in Figure 9 references a western in (D), which does not exist.
RESPONSE: Thank you for noticing, we corrected it.
Round 2
Reviewer 1 Report
The authors have addressed my concerns. I think this article should now be published.